# Emplacement of the Argyle diamond deposit into an ancient rift zone triggered by supercontinent breakup

Hugo K. H. Olierook [1,2] ✉, Denis Fougerouse [3], Luc S. Doucet [4,5], Yebo Liu [4], Murray J. Rayner[6], Martin Danišík [2], Daniel J. Condon [7], Brent I. A. McInnes[2], A. Lynton Jaques[8], Noreen J. Evans [2], Bradley J. McDonald[2], Zheng-Xiang Li[4,9], Christopher L. Kirkland [1], Celia Mayers[2] & Michael T. D. Wingate [2,10]

Argyle is the world's largest source of natural diamonds, yet one of only a few economic deposits hosted in a Paleoproterozoic orogen. The geodynamic triggers responsible for its alkaline ultramafic volcanic host are unknown. Here we show, using U-Pb and (U-Th)/He geochronology of detrital apatite and detrital zircon, and U-Pb dating of hydrothermal titanite, that emplacement of the Argyle lamproite is bracketed between $1311 \pm 9$ Ma and $1257 \pm 15$ Ma (2σ), older than previously known. To form the Argyle lamproite diatreme complex, emplacement was likely driven by lithospheric extension related to the breakup of the supercontinent Nuna. Extension facilitated production of low-degree partial melts and their migration through transcrustal corridors in the Paleoproterozoic Halls Creek Orogen, a rheologically-weak rift zone adjacent to the Kimberley Craton. Diamondiferous diatreme emplacement during (super)continental breakup may be prevalent but hitherto under-recognized in rift zones at the edges of ancient continental blocks.

The vast majority of primary economic diamond deposits are hosted in kimberlite diatremes located in Archean cratons[1,2], with characteristically old and thick continental lithosphere thought to be necessary for sustained diamond growth[3]. However, the largest source of natural diamonds discovered to date, the Argyle mine in the Kimberley region of Western Australia, is one of only a few economic deposits found within a Paleoproterozoic orogen adjacent to cratonic regions underlain by Archean basement[2,4]. Moreover, Argyle is hosted in olivine lamproite, rather than kimberlite, and has yielded >90% of all pink diamonds discovered[5]. The discovery of Argyle in 1979 resulted in a paradigm shift that led to diamond exploration in non-Archean terranes[6]. Despite its importance, the geodynamic drivers responsible for the emplacement of such an unusual diamondiferous pipe complex remain unclear.

The Argyle (formally AK1) lamproite is situated in the Carr Boyd Basin, a small Meso- to Neoproterozoic intracontinental basin in the Paleoproterozoic to Paleozoic Halls Creek Orogen at the southeastern edge of the Kimberley Craton[4]. Argyle is situated in an intracontinental rift that formed as a result of various, discrete far-field tectonic drivers between c. 1910 and 1805 Ma, the most important of which are the c.

[1]Timescales of Mineral Systems Group, School of Earth and Planetary Sciences, Curtin University, GPO Box U1987 Perth, WA 6845, Australia. [2]John de Laeter Centre, Curtin University, GPO Box U1987 Perth, WA 6845, Australia. [3]School of Earth and Planetary Sciences, Curtin University, GPO Box U1987 Perth, WA 6845, Australia. [4]Earth Dynamics Research Group, School of Earth and Planetary Sciences and The Institute for Geoscience Research (TIGeR), Curtin University, GPO Box U1987 Perth, WA 6845, Australia. [5]State Key Laboratory of Geological Processes and Mineral Resources, China University of Geosciences, Wuhan, Hubei 430074, China. [6]Rio Tinto, Perth 6000 Western Australia, Australia. [7]British Geological Survey, Keyworth, Nottingham NG12 5GG, UK. [8]Research School of Earth Sciences, Australian National University, Canberra ACT 2000, Australia. [9]Laoshan Laboratory, 266237 Qingdao, China. [10]Geological Survey of Western Australia, 100 Plain Street, East Perth, WA 6004, Australia. ✉e-mail: hugo.olierook@curtin.edu.au

1870–1850 Ma Hooper and c. 1835–1805 Ma Halls Creek Orogenies[7]. Argyle comprises four merged, NNE-striking diatremes compartmentalised by numerous NNW-striking faults[4], with the most diamond-rich diatreme (in excess of 20 carats per tonne[8]) towards the southern end of the deposit (Fig. 1B). Each diatreme comprises volcaniclastic olivine lamproite intruded by sparse narrow dikes of olivine lamproite. The predominant volcaniclastic lithologies are quartz-rich lapilli tuffs and coarse-grained ash tuffs ('sandy tuff'), comprising olivine lamproite lapilli mixed with variable amounts of quartz grains and country rock fragments. These lithologies formed by phreatomagmatic eruptions of the lamproite magma through unconsolidated water-rich sands and silts of the Carr Boyd Group in a maar-diatreme complex[4,9]. The northernmost diatreme is infilled with olivine lamproite tuffs free of detrital quartz grains ('non-sandy tuff'). The phreatomagmatic eruptions involving extensive interaction with water-saturated sediments were syn-eruptively accompanied by subsidence of the tephra, and by concomitant and pervasive hydrothermal alteration of the pipe[4,10]. Following emplacement, deformation of the Argyle pipe resulted in gentle tilting of the diatreme complex to the north[4]. Exhumation to its

present depth likely occurred in the Early Cretaceous as inferred from apatite fission track analysis on samples proximal to Argyle[11].

The age of emplacement of the Argyle lamproite has been estimated from whole-rock and altered phlogopite K-Ar and Rb-Sr dates, ranging from 1240 to 1110 Ma (see Supplementary Table 1[12–14]). However, K-Ar and Rb-Sr systematics can easily be modified by alteration[15], with previous workers recognising that the K contents of dated phlogopite in the main diatreme complex were "anomalously low" and indicative of partial chloritization[12]. The extensive alteration of K-bearing minerals means they are likely unsuitable as geochronometers for Argyle[16]. Although there is relatively fresh phlogopite in some of the lamproite dykes[17], dating it with high-precision $^{40}$Ar/$^{39}$Ar has proved challenging due to the presence of excess radiogenic Ar. Moreover, whilst the excess radiogenic Ar issue may be obviated with in situ Rb-Sr dating[18,19], minor alteration of the phlogopite and uncertainty around whether the dykes are coeval with the diatreme still make dating these phlogopite grains problematic.

Here, we provide a combination of petrographic observations, U-Pb geochronology of titanite, and U-Pb and (U-Th)/He dating of zircon and apatite separated from a sample of high grade drill core of the Argyle lamproite to resolve the age of the Argyle deposit and provide an explanation for when and why Argyle formed.

## Results
### Petrography of the Argyle lamproite
A representative drill core sample was provided by Rio Tinto from the high-grade mineralised diatreme (AK1-Lh01), with two thin sections prepared from this material. AK1-Lh01 comprises ~45% lamproite clasts (chiefly chlorite and serpentine after original olivine and phlogopite), ~40% quartz clasts, ~10% calcite cement, and 4% late-stage titanite that rims combined quartz–lamproite clusters (Fig. 2, all vol. %; Supplementary Table 2 and Supplementary Fig. 1). Except for the largest clasts, quartz clasts are well rounded with no evidence of authigenic quartz overgrowths, indicative of a detrital origin with minimal diagenetic overprint. Titanite forms a quasi-connected network surrounding combined lamproite-detritus clasts (Fig. 2B). Titanite is anhedral and contains inclusions of the lamproite clasts (chlorite and serpentine), detrital matrix (quartz, zircon and apatite) and calcite cement (Fig. 2D). This petrographic evidence indicates that titanite post-dates emplacement of the lamproite and carbonate alteration (Fig. 2B–D). The titanite was likely formed by dissolving original perovskite from the lamproite with the addition of silica from the surrounding maar-derived meteoric waters[17], implying that the titanite was a product of deuteric alteration and only very shortly post-dates lamproite emplacement, well within uncertainty of any radiometric technique. Trace rounded zircon and subangular apatite are also present within the detrital matrix. Although not present in thin section, diamonds were also found through disaggregation of the core sample (e.g., ~1 mm dodecahedral diamond fragment; Supplementary Fig. 2).

### Age for the Argyle (AK1) lamproite diamond deposit
Three minerals—detrital zircon and detrital apatite, and hydrothermal titanite—were dated using U-Pb laser ablation inductively coupled plasma mass spectrometry (LA-ICP-MS) to provide a maximum depositional age (apatite, zircon) and a minimum age of hydrothermal alteration (titanite), which together bracket emplacement of the Argyle lamproite (see Supplementary Fig. 3 and Supplementary Data 1–2). Selected titanite were re-dated using U-Pb isotope dilution thermal ionisation mass spectrometry (ID-TIMS) to aid in improving the precision. Detrital zircon and apatite were also dated using (U-Th)/He to understand the low-temperature history at Argyle, and to test whether hydrothermal titanite formed after exhumation.

Detrital apatite yields a slightly overdispersed ($p < 0.05$) spread of $^{207}$Pb-corrected dates centred at c. 1900–1820 Ma (Fig. 3A, B). Coupled with the angular nature of the apatite grains (Supplementary Fig. 3),

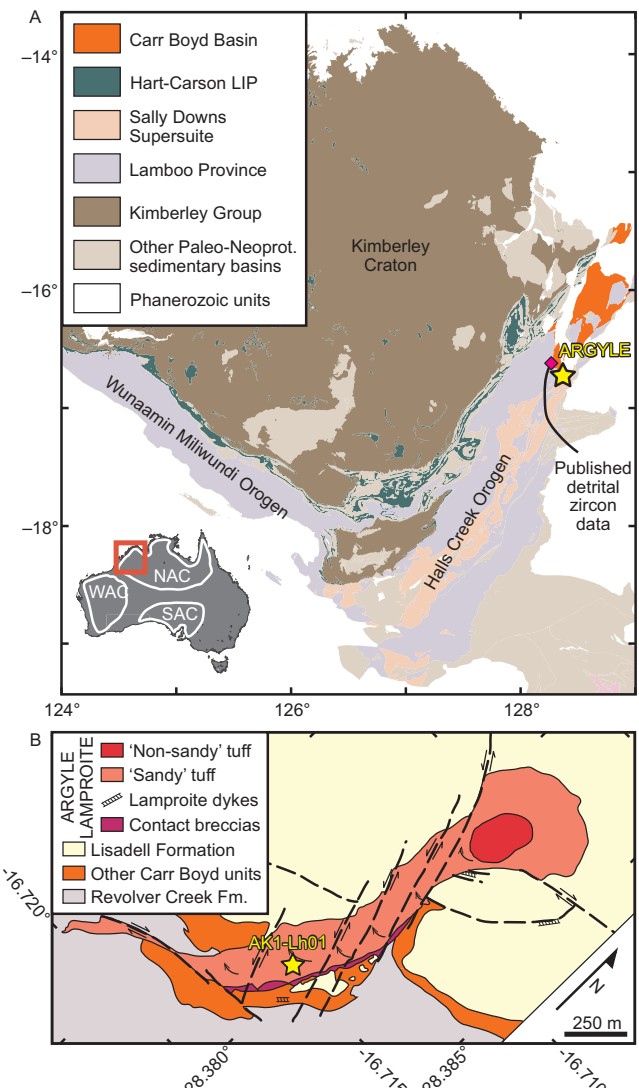

**Fig. 1 | Geological setting of the Argyle (AK1) lamproite. A** 1:500,000 geological map of the Kimberley Craton and surrounding orogens, with inset showing the location of the West (WAC), North (NAC) and South (SAC) Australian Cratons. **B** Simplified geological map of the Argyle lamproite with approximate sample location.

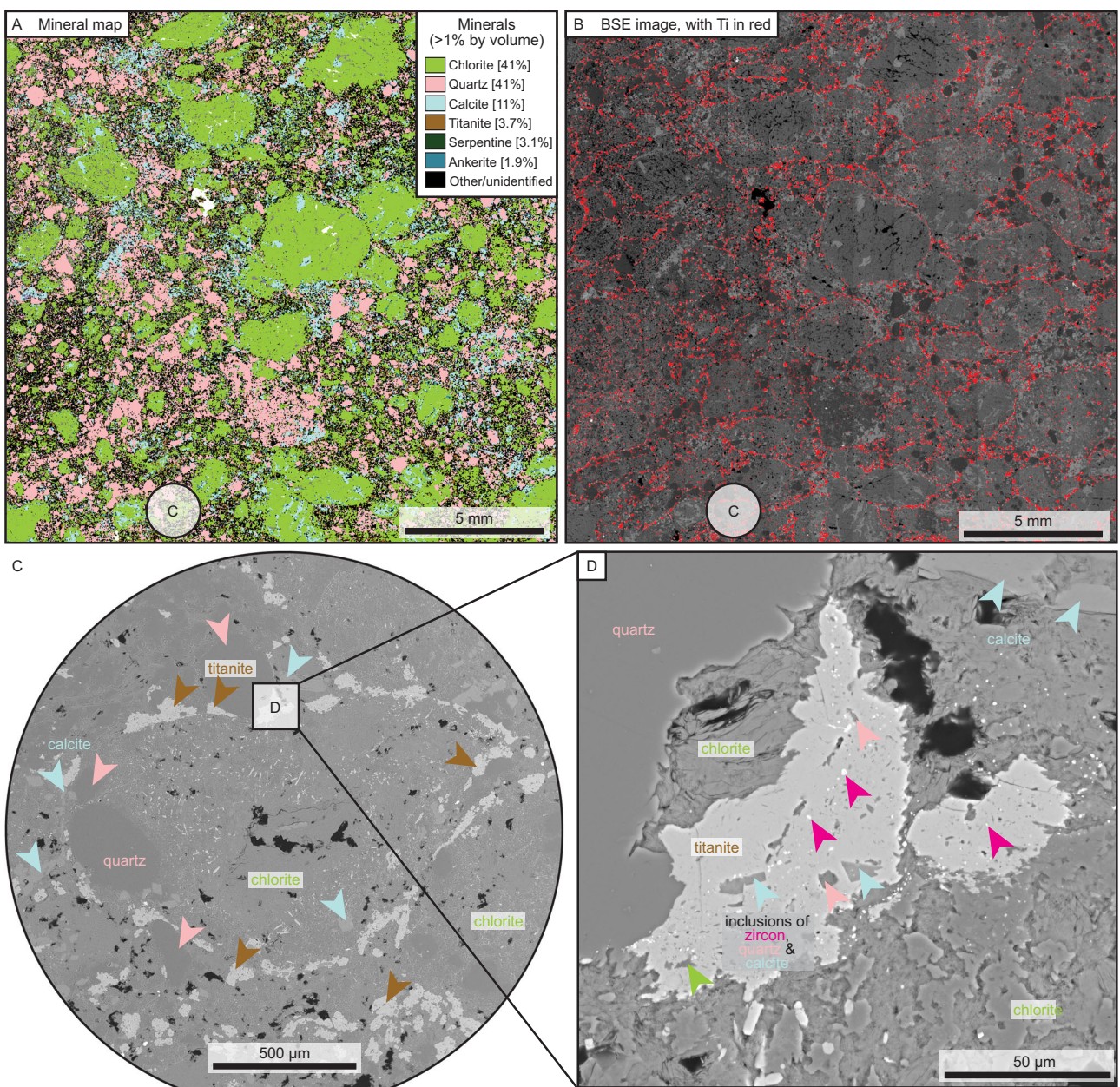

**Fig. 2 | Mineral association and petrography of high-grade sample AK1-Lh01a at Argyle. A** Automated mineral analysis map of major and minor minerals. **B** Backscattered electron map of **A** but showing Ti Kα peak overlain in red, the vast majority of which corresponds to titanite. **C** Expanded view of **B**, showing late-stage titanite rimming a cluster of lamproite, detrital quartz and calcite. **D** Expanded view of **C**, showing inclusions of zircon, quartz and calcite within titanite. For additional petrographic images, see Supplementary Table 2 and Supplementary Fig. 1.

slight dispersion in the data can be explained by the apatite detritus originating from erosion of proximal rocks formed during the c. 1870–1850 Ma Hooper and c. 1835–1805 Ma Halls Creek orogenies, with a few older grains (Fig. 3A). As none of the apatite analyses is concordant, individual [207]Pb-corrected dates are model ages, dependent on an assumed common-Pb composition and closed system behaviour. The youngest coherent group of detrital apatite [207]Pb-corrected dates yields a weighted mean of $1828 \pm 6$ Ma (MSWD = 1.01, $p = 0.40$) that, whilst providing a maximum emplacement age of the Argyle lamproite, is significantly older than the Meso- to Neoproterozoic Carr Boyd Group in which Argyle is hosted. Importantly, the U-Pb system in these apatite grains was not thermally reset (i.e., heated above ~450 °C, which is the closure temperature to Pb in apatite[20]), as after the c. 1835–1805 Ma Halls Creek Orogeny there is no known local

tectonothermal event of sufficient magnitude to cause open system behaviour.

Detrital zircon analyses define a major peak at c. 1870 Ma, implying derivation from rocks formed during the c. 1870–1850 Ma Hooper Orogeny, as well as minor peaks at c. 2.5 Ga, 2.0 Ga, c. 1.8 Ga and c. 1.6 Ga (amongst even rarer age components; Fig. 3B, C). This age spectrum is similar to other detrital zircon age spectra from the Carr Boyd Group sedimentary rocks into which Argyle was emplaced, although the published data have a higher proportion of c. 1835–1805 Ma detrius derived from rocks crystallised during the Halls Creek Orogeny (Fig. 3B[21,22]). The most important subordinate detrital zircon ages are three analyses younger than 1360 Ma, with the youngest two of these yielding a [206]Pb/[238]U weighted mean age of $1311 \pm 9$ Ma (MSWD = 1.02, $p = 0.31$; Fig. 3C), providing a robust maximum

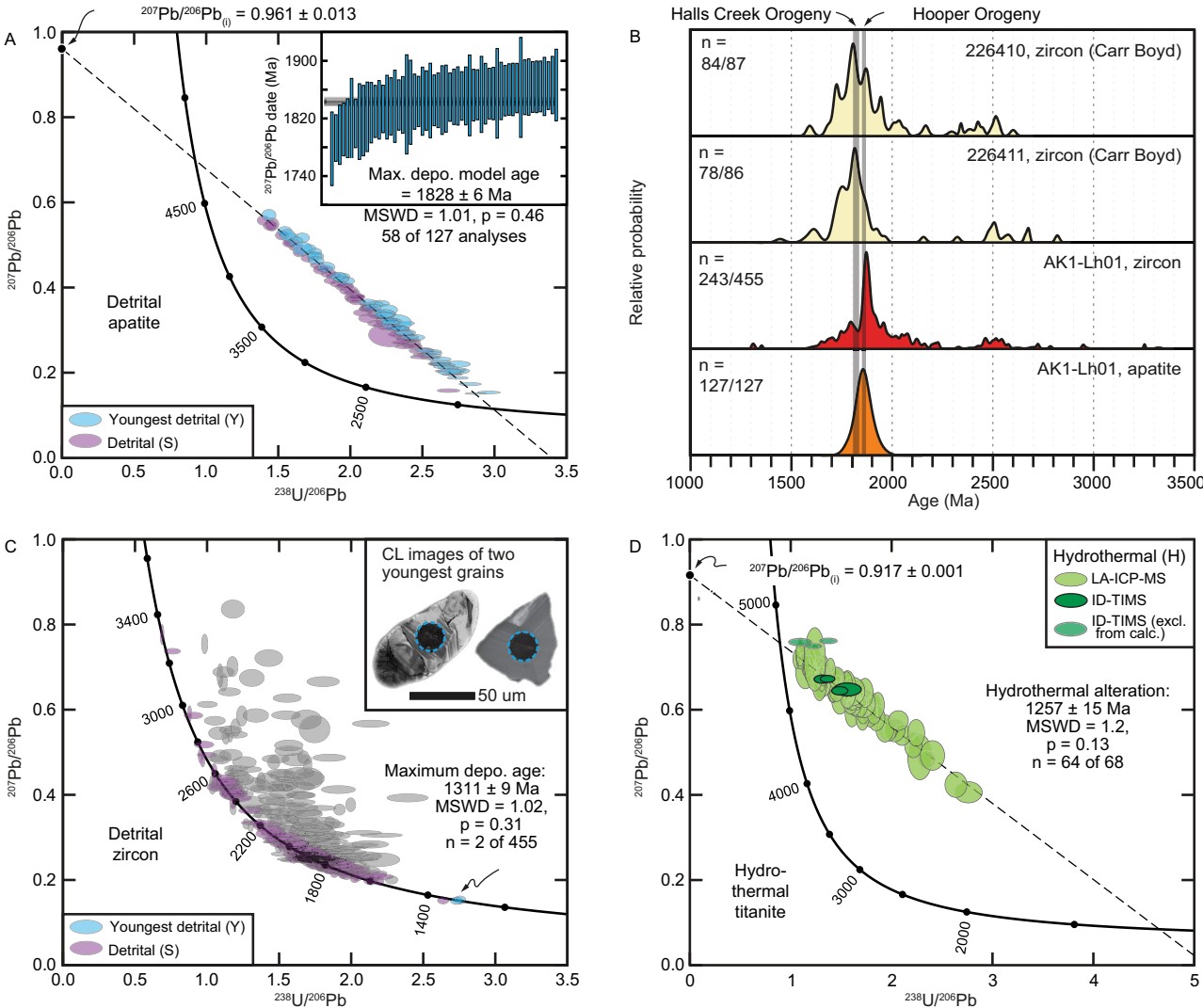

**Fig. 3 | Geochronological data from Argyle sample AK01-Lh01. A** Detrital apatite concordia plot; inset shows the weighted mean of youngest grains. **B** Probability density plot of detrital zircon and apatite, including published zircon data from two proximal samples from elsewhere in the Carr Boyd Basin[21,22]. **C** Detrital zircon, with inset showing cathodoluminescence images of the two youngest grains. **D** Hydrothermal titanite concordia plot.

emplacement age for the Argyle lamproite (see Supplementary Fig. 4 for evaluation of data quality).

Titanite from Argyle is small (<50 μm, necessitating a small laser spot size of 30 μm), low in U (5–25 ppm) and rich in common Pb (38–78%), consistent with a hydrothermal origin[23], but meaning that only low-precision dates can be obtained using LA-ICP-MS. Free-regression through all titanite data collected using LA-ICP-MS yields a robust but imprecise lower concordia intercept of 1268 ± 40 Ma (MSWD = 1.3, p = 0.09, n = 58, Fig. 3D). Ten selected grains were re-dated using ID-TIMS, which broadly show similar trends as LA-ICP-MS data but with excess scatter to the right of the discordia. Potential causes for this scatter include recent Pb-loss through alteration or ancient Pb-loss during a subsequent tecto-nothermal event. Using only the six analyses that broadly overlap with the LA-ICP-MS data yields a free-regressed lower concordia intercept of 1271 ± 40 Ma (MSWD = 0.75, p = 0.56, n = 6 of 10, Fig. 3D), very similar in both precision and accuracy due to the higher precision of individual analyses but a lower spread along discordia. Harnessing both the spread of the LA-ICP-MS data and the precision of the ID-TIMS data, we compute a combined free-regressed age of 1257 ± 15 Ma (MSWD = 1.2, p = 0.13, n = 64 of 68, Fig. 3D). Importantly, as titanite has a higher closure temperature to

Pb (~650 °C[24]) than apatite (~450 °C[20]), this c. 1260 Ma age cannot reflect a discrete heating and cooling event, as detrital apatite would also be reset. Thus, the age of 1257 ± 15 Ma reflects the pervasive hydrothermal alteration of the combined lamproite–detritus clasts. Given that there is mild scatter on the ID-TIMS and that this may be present but masked by the higher uncertainties in the LA-ICP-MS, the age of 1257 ± 15 Ma records in a most conservative sense a minimum emplacement age for the Argyle diatreme complex.

If exhumation occurred prior to hydrothermal titanite formation, apatite and zircon (U-Th)/He (with closure temperatures to He of ~75 °C and ~180 °C, respectively[25]) could provide better age constraints on lamproite eruption. Seven of eight detrital apatite grains yielded a corrected (U-Th)/He weighted mean age of 121 ± 10 Ma (MSWD = 1.4, p = 0.21; Supplementary Fig. 5), which can be readily linked to Cre-taceous exhumation of the Kimberley Craton[11]. Detrital zircon crystals yielded overdispersed (U-Th)/He dates ranging from c. 1150 to 185 Ma, which could relate to passage through the partial He retention zone. The oldest dates provide a minimum age for the lamproite (Supplementary Fig. 5), although this postdates hydrothermal titanite formation.

In summary, the best age constraints for emplacement of the Argyle lamproite are provided by a maximum depositional zircon age

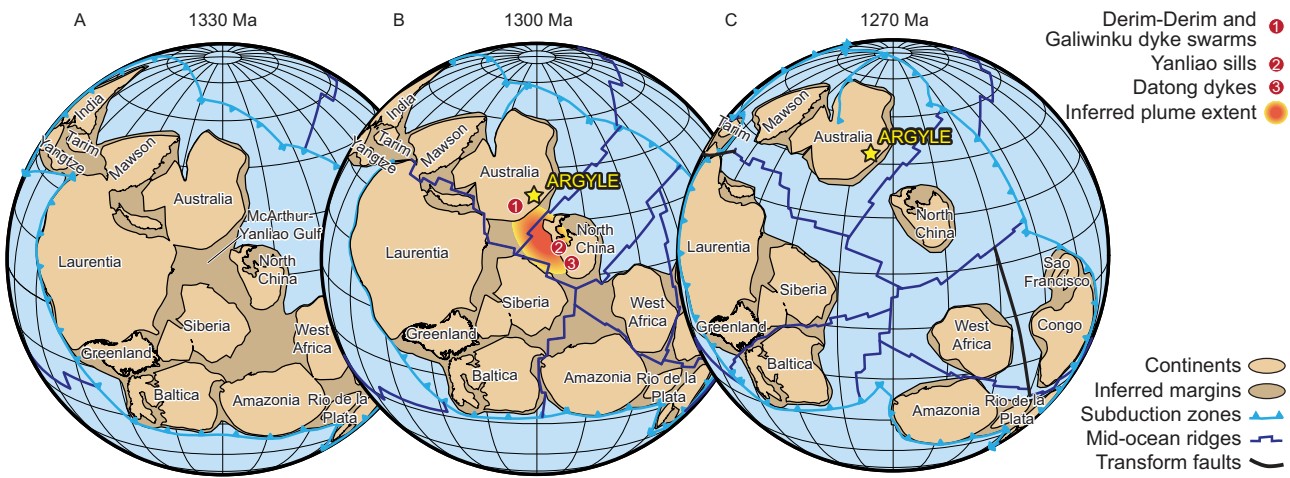

**Fig. 4 | Plate tectonic reconstructions showing the position of Argyle within the McArthur-Yanliao Gulf. A** At 1330 Ma. **B** At 1300 Ma. **C** At 1270 Ma. The paleo-geographic reconstruction is made using GPlates 2.3 open-source software (https://www.gplates.org/) with the full-plate model of Li et al.[30]. The potential extent of plume head that caused the Derim Derim-Galiwinku-Yanliao LIP is also shown, with its dimension elongated along rift zones[28, 34].

of $1311 \pm 9$ Ma and a minimum hydrothermal titanite alteration age of $1257 \pm 15$ Ma.

## Discussion

At c. 1300 Ma, the supercontinent Nuna was on the cusp of breakup[26]. At the time, most plate tectonic reconstructions place Argyle at the periphery of the McArthur-Yanliao Gulf (Fig. 4), an intercontinental Mesoproterozoic basin between proto-Australia, North China, Siberia and Laurentia[27–32] (and in some reconstructions also including India[29,31] and/or South China[28,29]). Temporally-constrained paleomagnetic and geological evidence indicates that Nuna was together at c. 1320 but dispersed by c. 1220 Ma, with scarce high-quality paleomagnetic data in the intervening 100 Myr making it difficult to be more precise on the exact timing of breakup[28].

At c. 1330–1295 Ma, dyke swarms and related volcanism emplaced around the periphery of the McArthur-Yanliao Gulf, covering part of the North China Craton (Yanliao rift zone, including the Yanliao sills and Datong dykes[33]) and the North Australian Craton (Derim Derim-Galiwinku dyke swarms[34]), which are collectively termed the Derim Derim-Galiwinku-Yanliao large igneous province (Fig. 4[34]). Given the paucity of known c. 1310 Ma source rocks in the vicinity of the Kimberley Craton, it is likely that these rare, young zircon grains found in Argyle were derived from the eroded portions of the Derim Derim-Galiwinku-Yanliao large igneous province. Such an interpretation is consistent with plate reconstructions[28], which would favour a sedimentary transport pathway from the centre of the McArthur-Yanliao Gulf towards a circum-Nuna ocean towards the northeast, draining past the Kimberley Craton and perhaps into the Halls Creek Orogen (Fig. 4).

Argyle is situated in the Halls Creek Orogen, a rheologically-weak rift zone with thinner lithosphere that is prone to reactivation[7]. Additional heat from the Derim Derim-Galiwinku-Yanliao large igneous province and/or mechanical extension from the breakup of Nuna could have reactivated mantle-to-crust pathways to facilitate rapid ascent of low-degree volatile-rich partial melts (e.g., lamproites) to the surface. The two mechanisms may also be intimately inter-related, with large igneous provinces commonly heralding continental breakup[35] and thermal weakening lowering the extensional forces required to initiate continental breakup[36,37]. Although the precise timing of the onset of Nuna breakup is not yet known, paleomagnetic constraints indicate that the West–North Australian cratons, North China Craton

and Laurentia were separated sometime between 1320 and 1220 Ma[28], coincident with emplacement of the Argyle lamproite.

Lithospheric extension in, or adjacent to, Archean cratons during continental rifting has previously been recognized as a mechanism for emplacement of kimberlites, lamproites and other related volatile- and incompatible element-enriched ultramafic diatremes[38–42]. Examples include the East European Platform during Nuna breakup (c. 1250–1230 Ma[43]), the Kimberley Craton during attempted Rodinia breakup (c. 840–800 Ma[44]) and the Kaapvaal Craton during Gondwana/Pangea breakup (c. 200–120 Ma[45,46]), amongst many others. However, the role of (super)continental breakup has only recently begun to be significantly appreciated for kimberlite emplacement during periods of continental breakup[2,38,42]. In all breakup-related cases, diatreme emplacement appears to be contemporaneous with breakup, but with a peak production of volatile-rich ultramafic rocks lagging behind the breakup period[41,42]. The same applies to Argyle and Nuna breakup. Whilst Argyle appears to be coincident with the onset of Nuna breakup, together with temporally near-coincident magmatism at Helpmekaar in South Africa[41] and kimberlites in the East European platform[43], it is not until 1200–1100 Ma that diatreme emplacement peaked worldwide, coincident with accelerated plate velocities[30]. Thus, there appears to be a first-order control of continental extension and plate velocities on (diamondiferous) kimberlite and related volatile-rich ultramafic diatreme production[41].

The global volatile-rich ultramafic diatreme record is undoubtedly affected by preservation bias, particularly for pre-Pangean diatremes, but it is second order to continental rifting processes. Continental amalgamation and associated post-orogenic relaxation/extension is known to be better at preserving the geological record[47,48]. Thus, one might expect that preservation would be higher at c. 1800–1600 Ma (Nuna assembly), 1100–900 Ma (Rodinia assembly), 650–500 Ma (Gondwana assembly) and 250–200 Ma (Pangea assembly)[30]. With the exception of Gondwana assembly, periods of continental breakup after supercontinent assembly are marked by significantly more kimberlite and related volatile-rich ultramafic production[38,49], further supporting the notion that crustal extension is a first order control that is not erased by preservation bias. The implication is that continental extension is one of the dominant driving forces in ensuring that volatiles are destabilised in the deep mantle and propagated towards the surface. That being said, there was still likely significant erosion of breakup-related diatremes, and it is perhaps for this reason that

diatremes concomitant with initial continental breakup are rare in the geological record. One only has to look at Argyle to appreciate the fortuity involved in preserving a diamond-bearing diatreme. It was buried to >5–6 km (>180 °C, zircon [U-Th]/He) and exhumed to <2 km by the Early Cretaceous, to only be partly eroded and at mineable depth at the present.

## Methods

### Thin section petrographic methods and preparation
Two polished thin sections were prepared from one sample, AK1-Lh01a and AK1-Lh01b, by Adelaide Petrographic Laboratories. Entire thin sections were imaged using an Axio Imager II in the School of Earth and Planetary Sciences, Curtin University, in transmitted plane-polarised, transmitted cross-polarised and reflected plane-polarised light to provide clear petrographic overviews.

The thin section was subsequently carbon coated and analysed using a Tescan Integrated Mineral Analyser (TIMA) in the John de Laeter Centre (JdLC), Curtin University, to aid in mineral identification, particularly with respect to identifying dateable minerals[50–52]. A TIMA is a field emission gun scanning electron microscope (FEG-SEM), equipped with four electron dispersive X-ray spectrometers (EDS), and capable of recording 420k X-ray counts per second. Thin sections were analysed in 'dot-mapping' mode with a rectangular mesh at a step-size of 3 μm for back-scattered electron (BSE) imaging. One thousand EDS counts are collected every 9th step (i.e., 27 μm) or when the BSE contrast changes (i.e., a change in mineral phase). For a given mineral grain, EDS counts are integrated across the entire grain. TIMA analyses used an accelerating voltage of 25 kV, a probe current of 5.5 nA, a spot size of 80 nm and a nominal working distance of 15 mm. After imaging and EDS collection, BSE signals and EDS peaks are referenced to a mineral library for automatic mineral classification.

Detailed backscattered electron (BSE) and cathodoluminescence (CL) imaging were undertaken on a Clara FEG-SEM in the JdLC using an accelerating voltage of 20 kV (BSE) or 12 kV (CL), a nominal working distance of 10.0 mm (BSE) or 16.5 mm (CL), and a beam intensity of 1 nA for both.

A full compendium of optical and scanning electron images for thin sections may be found in Supplementary Fig. 1.

### Geochronological processing and grain imaging analysis
Approximately 2 kg of sample AK1-Lh01 was cut into ~2–3 cm cubes and disaggregated using a SelFrag high-voltage pulse fragmentation system in the JdLC, Curtin University, to liberate constituent minerals. Resultant slurries were run through a Jasper Canyon Research concentrating shaker table for initial concentration of heavy minerals and subsequently through lithium heteropolytungstate (LST) heavy liquids at 2.9 g cm$^{-3}$ and through methylene iodide at 3.3 g cm$^{-3}$. Magnetic minerals were removed from the >3.3 g cm$^{-3}$ and 2.9–3.3 g cm$^{-3}$ fractions using a hand magnet and a Frantz isodynamic magnetic separator to concentrate zircon, apatite and titanite. Both sets of concentrates were subsequently dump-mounted in 25 mm epoxy rounds. All mounts were polished to half-grain thickness to expose grain interiors. Mounted grains were imaged with transmitted and reflected light on an optical microscope and, subsequently, CL imaged (zircon) and BSE imaged (titanite and apatite). Transmitted and reflected light images were used to assess grain shape and transparency. CL and BSE images were used to document internal zonation patterns (e.g. oscillatory, sector), recognise inclusions and identify growth and recrystallisation textures[53]. A full compendium of CL and BSE images may be found in Supplementary Fig. 3. U-Pb spots were not pre-selected to mitigate bias in detrital populations[54].

We also note that, during a short examination of the coarse (>425 μm) mineral fraction under a binocular microscope, one sub-hedral diamond was discovered in AK1-Lh01 (Supplementary Fig. 2).

### Detrital apatite U-Pb dating via laser ablation inductively coupled plasma mass spectrometry (LA-ICP-MS)
Apatite U-Pb isotope measurements were collected at the GeoHistory Facility, JdLC, Curtin University. An excimer laser (Resonetics S-155-SE 193 nm) was used at a spot size of 50 μm, on-sample fluence of 2.2 J cm$^{-2}$ and repetition rate of 6 Hz. Ablations on reference materials and unknowns were undertaken over 30 s with 40 s of background capture. All analyses were preceded by two cleaning pulses. The sample cell was flushed by ultrahigh purity He (0.32 L min$^{-1}$) and N$_2$ (1.2 mL min$^{-1}$).

Apatite U-Pb data were collected on an Agilent 8900 triple quadrupole mass spectrometer with high-purity Ar as the carrier gas for both sessions (flow rate 0.98 L min$^{-1}$). Analyses of ~20 unknowns were bracketed by analysis of a standard block containing the primary apatite reference material Mount McClure (523.5 ± 1.5 Ma; ref. [55]), which was used to monitor and correct for mass fractionation and instrumental drift. The standard block also contained a range of secondary apatite standards, including Durango (31.44 ± 0.18 Ma; ref. [56]), MAD2 (474.25 ± 0.41 Ma; ref. [57]) and AS3/FC-Duluth (1099.1 ± 1.2 Ma; ref. [58]), which were used to monitor data accuracy and precision. The secondary standards and unknowns were reduced against Mount McClure using regressions anchored to contemporaneous common $^{207}$Pb/$^{206}$Pb values from the reference material. During the analytical session, Durango, MAD2 and FC-Duluth yielded statistically-reliable ($p > 0.05$) regressed ages of 29 ± 7 Ma, 476 ± 13 and 1091 ± 40 Ma (all uncertainties at 2 s.d.), respectively, all of which are within uncertainty of the published age (see Supplementary Data 1 for full U-Pb reference material compilation).

Data were reduced using the VizualAge UComPbine data reduction scheme in Iolite4[59,60]. Uncertainties on analyses of primary reference materials were propagated in quadrature to the unknowns and secondary apatite reference materials. For the unknowns, some analyses ablated across inadvertent inclusions; these inclusion-related signals were cropped from the integrations; although this maintains the accuracy of the cropped analyses, the U-Pb precision is significantly reduced due to the shorter integration times[51,61]. If inclusions were too large or numerous, the integrations were deleted. No corrections were applied for common Pb because almost all analyses have appreciable amounts of common Pb (Supplementary Data 2). Age calculations and plots were made using Isoplot 4.15[62]. $^{207}$Pb-corrected $^{206}$Pb/$^{238}$U model dates were computed for each individual grain using the $^{207}$Pb/$^{206}$Pb$_{(i)}$ of a free-regressed line through the data. However, we caution that, due to slight excess scatter (MSWD = 2.5) and potential unrecognised Pb loss or incorporated common Pb of alternative $^{207}$Pb/$^{206}$Pb ratios, these $^{207}$Pb-corrected $^{206}$Pb/$^{238}$U dates should strictly be treated as model ages. All spot analyses are presented at 2σ and weighted mean analyses are presented at 95% confidence.

Full isotopic data for the reference materials and samples are given in Supplementary Data 1 and 2, respectively.

### Detrital zircon U-Pb dating via LA-ICP-MS
Zircon U-Pb measurements were collected across two sessions at the GeoHistory Facility, JdLC, Curtin University. The second session was necessitated as only a single analysis at c. 1310 Ma was recognized in the first session. For both sessions, a RESOlution SE 193 nm ArF with a Lauren Technic S155 cell was used. The beam diameter was 30 μm, on-sample energy was 3.2 J cm$^{-2}$ with a repetition rate of 6 Hz for 25 s of analysis time and ~45 s of background capture. All analyses were preceded by two cleaning pulses. The sample cell was flushed by ultrahigh purity He (0.32 L min$^{-1}$) and N$_2$ (1.2 mL min$^{-1}$).

Zircon U-Pb isotope data were collected on an Agilent 8900 triple quadrupole mass spectrometer for all sessions with high purity Ar as the carrier gas (flow rate 0.98 L min$^{-1}$). Analyses of ~20 unknowns were bracketed by analysis of a standard block containing the primary zircon reference materials GJ-1 (601.95 ± 0.40 Ma; refs. [63,64]) and, for the

second session, OG1 (3465.4 ± 0.6 Ma; ref. 65), which were used to monitor and correct for mass fractionation and instrumental drift. The standard block also contained the Phanerozoic to Archean reference materials Rak-17 (first session only, 295.56 ± 0.21 Ma; International Association of Geoanalysts, pers. comms. 2019), Plešovice (337.13 ± 0.37 Ma; ref. 66), 91500 (1063.78 ± 0.65 Ma; refs. 64,67), R33 first session only, 418.9 ± 0.4 Ma; ref. 68), FC-1 (first session only, 1099.0 ± 0.6 Ma; ref. 69) and Maniitsoq (3008.70 ± 0.72 Ma; ref. 70; all uncertainties at 2σ), which were used to monitor data accuracy and precision. During the analytical sessions, when reduced against a matrix-matched reference material, Rak-17, Plešovice, R33, 91500, FC-1 and Maniitsoq yielded statistically-reliable ($p > 0.05$) weighted mean ages of 293 ± 6 Ma, 340.1 ± 3.8 Ma to 340.5 ± 3.4 Ma, 417 ± 7 Ma, 1059 ± 11 Ma to 1063 ± 14, 1101 ± 13, and 3010 ± 36 to 3013 ± 17 Ma (all uncertainties at 2 s.d.), respectively, all of which are within the published age (see Supplementary Data 1 for full reference material compilation).

U-Pb isotopic data were reduced in Iolite4[59]. Uncertainties on analyses of primary reference materials were propagated in quadrature to the unknowns and secondary zircon reference materials. For the unknowns, some analyses ablated across inadvertent inclusions; these inclusion-relates signals were cropped from the integrations or, if too large or numerous, integrations deleted; whilst the accuracy of the cropped analyses is maintained, the precision of the U-Pb data is significantly reduced due to the shorter integration times[71–73]. No corrections were applied for common Pb because common Pb was below detection limits for almost all concordant analyses (f206, Supplementary Data 2). Zircon analyses are considered concordant where the error ellipses at 2σ generated by the $^{207}Pb/^{206}Pb$ and $^{206}Pb/^{238}U$ ratios overlap the inverse concordia curve, excluding uncertainties on the decay constant. Age calculations and plots were made using Isoplot 4.15[62]. All zircon dates >1.5 Ga are presented as $^{207}Pb/^{206}Pb$ ages and those <1.5 Ga are presented as $^{206}Pb/^{238}U$ ages for optimum precision[74]. All spot analyses are presented at 2σ and weighted mean analyses are presented at 95% confidence.

Full isotopic data for the reference materials and samples are given in Supplementary Data 1 and 2, respectively.

### Hydrothermal titanite U-Pb dating via LA-ICP-MS

Titanite U–Pb isotope data were collected at the GeoHistory Facility, JdLC, Curtin University. An excimer laser (Resonetics S-155-SE 193 nm) was used with a spot size of 30 μm, an on-sample fluence of 2.2 J cm$^{-2}$ and repetition rate of 5 Hz for ~25 s of total analysis time and 40 s of background capture. All analyses were preceded by two cleaning pulses. The sample cell was flushed by ultrahigh purity He (0.32 L min$^{-1}$) and N$_2$ (1.2 mL min$^{-1}$).

U–Pb data were collected on an Agilent 8900 triple quadrupole mass spectrometer with high purity Ar as the carrier gas for both sessions (flow rate 0.98 L min$^{-1}$). Analyses of ~20 unknowns were bracketed by analysis of a standard block containing the primary titanite reference materials MKED 1 (1517.32 ± 0.32 Ma; ref. 75), which was used to monitor and correct for mass fractionation and instrumental drift. The standard block also contained the Phanerozoic to Proterozoic reference materials Khan (522.2 ± 2.2 Ma; ref. 76) and BLR-1 (1047.1 ± 0.4 Ma; ref. 77), which were used to monitor data accuracy and precision. The secondary standards and unknowns were reduced against MKED1 using regressions anchored to contemporaneous common $^{207}Pb/^{206}Pb$ values from the reference material. During the analytical session, Khan and BLR-1 yielded statistically-reliable ($p > 0.05$) regressed ages of 529 ± 8 Ma and 1050 ± 16 Ma (all uncertainties at 2 s.d.), both of which are within uncertainty of the published age (see Supplementary Data 1 for full compilation of results for U-Pb reference materials).

Data were reduced using U-Pb geochronology in Iolite4[59,60]. Uncertainties on the primary reference materials were propagated in

quadrature to the unknowns and secondary titanite reference materials. For the unknowns, some analyses inadvertently ablated through inclusions represented by short time duration spikes in characteristic isotopes (e.g., $^{232}Th$ for monazite); these were cropped from the sample integration or, if too large or numerous, that sample integration was deleted. While the accuracy of the cropped analyses is maintained, the precision of the U-Pb is significantly reduced due to the shorter integration time[23,78,79]. No corrections were applied for common-Pb was not corrected because almost all analyses have appreciable amounts of common Pb (Supplementary Data 2). Age calculations and plots were made using Isoplot 4.15[62]. $^{207}Pb$-corrected $^{206}Pb/^{238}U$ model dates were computed for each individual grain using the $^{207}Pb/^{206}Pb_{(i)}$ of a free-regressed line through the data. All spot analyses are presented at 2σ and weighted mean analyses are presented at 95% confidence.

Full isotopic data for the reference materials and samples are given in Supplementary Data 1 and 2, respectively.

### Hydrothermal titanite U-Pb dating via ID-TIMS

Selected titanite crystals that had previously been ablated and analysed with ICP-MS were extracted from a resin mount at the British Geological Survey, UK. Ten grains were plucked from the mount used for LA-ICP-MS. Eight of these grains came from one area of the mount, with the other two grains selected based on different ratios of radiogenic to common Pb (as per the LA-ICP-MS discordia in Fig. 3D). All of the analysed titanite crystals were ultrasonically cleaned for an hour before being placed on a hotplate for 30 min, and rinsed in ultrapure acetone. After rinsing, titanite fractions were transferred to 300 μl Teflon PFA microcapsules, and leached in ~120 μl of 29 M HF with a trace amount of 4 M HNO$_3$ for 12 h at 180 °C. The acid solution was then removed, and titanite crystals were rinsed again in 4 M HNO$_3$ and 6 M HCl before spiking with mixed EARTHTIME $^{233}U–^{235}U–^{205}Pb$ tracer (ET535[80,81]), placed in Parr vessels at ~220 °C for 60 h, dried to fluorides and then converted to chlorides by mixing in 3 M HCl at ~180 °C overnight. U and Pb were separated using standard HBr and HNO$_3$-based anion-exchange chromatographic procedures on 0.05 ml PRFE columns. Isotopic ratios were measured at the British Geological Survey using a Thermo Scientific Triton Thermal Ionisation Mass-Spectrometer (TIMS). Pb and U were loaded separately on single Re filaments in a silica-gel/phosphoric acid mixture. Pb and U were measured by peak hopping on a single SEM detector, with Pb isotopes corrected for mass bias using a fractionation factor of 0.12 ± 0.04%/ amu (1σ) and U mass fractionation was calculated in real time based on the isotopic composition of the ET535 tracer. Oxide correction was based on an $^{18}O/^{16}O$ ratio of 0.00205 ± 0.00004, and the sample $^{238}U/^{235}U$ ratio was assumed to be 137.818 ± 0.045[82]. Data reduction was based upon the algorithms of Schmitz and Schoene[83].

### Apatite and zircon (U-Th)/He dating

(U-Th)/He dating of apatite and zircon was conducted at the Western Australia ThermoChronology Hub Facility, JdLC, Curtin University, and followed the procedures detailed in Danišík, Štěpančíková[84,] Danišík, McInnes[85,] and Danišík, Lowe[86]. Single crystals of apatite and zircon were hand-picked following strict selection criteria with regard to their morphology, clarity, and presence of inclusions, then photographed and measured for physical dimensions in order to calculate alpha-ejection correction factors[87]. Selected crystals were loaded in Pt (apatite) or Nb (zircon) tubes and loaded into an Alphachron II instrument for He extraction. Together with other gases, ⁴He was extracted at ~960 °C (apatite) or ~1250 °C (zircon) under ultra-high vacuum using a diode laser, cleaned on Ti-Zr getters, and spiked with 99.9% pure ³He gas. The volume of ⁴He was measured by isotope dilution on a QMG 220 M1 Pfeiffer Prisma Plus mass spectrometer. A 're-extract' was run after each analysis to verify complete outgassing of the crystal. Helium gas signals were corrected for blank, determined by

analysing empty Nb or Pt microtubes interspersed between the unknowns using the same gas extraction procedure.

After the He measurements, microtubes containing the crystals were retrieved from the Alphachron, spiked with $^{235}$U and $^{230}$Th, and dissolved. Apatite was dissolved in $HNO_3$ at room temperature using ultrasonication; zircon was dissolved in Parr acid digestion vessels in two cycles of HF, $HNO_3$ (cycle 1), and HCl acids (cycle 2) following the procedures described in Evans, Byrne[88]. Sample, blank, and spiked standard solutions were then diluted by Milli-Q water and analysed by isotope dilution for $^{238}$U and $^{232}$Th, and by external calibration for $^{147}$Sm on an Element XR™ High Resolution ICP-MS. The total analytical uncertainty (TAU) was calculated by addition in quadrature on He and weighted uncertainties on U, Th, and Sm measurements. The (U–Th)/ He ages were corrected for α-ejection (Ft correction) after Farley, Wolf[87], whereby a homogeneous distribution of U, Th, and Sm was assumed for the crystals. Accuracy of the (U–Th)/He dating procedure was monitored by replicate analyses of Fish Canyon Tuff zircon ($n = 3$) and Durango apatite ($n = 4$) measured over the period of this study as internal standards, yielding mean (U–Th)/He ages of $28.5 \pm 3.1$ Ma and $31.5 \pm 1.9$ Ma, respectively (2 s.d.). These ages are in good agreement with the reference material ages of $28.3 \pm 1.3$ Ma (Fish Canyon Tuff[89]) and $31.13 \pm 1.01$ Ma (Durango[56]).

## Data availability

All data used in this manuscript are included in the Supplementary Information File and Supplementary Data Files 1 and 2.

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

## Acknowledgements

Funding was provided by the Geological Survey of Western Australia (GSWA) to D. F. and the open fund of State Key Laboratory of Geological Processes and Mineral Resources, China University of Geosciences, Wuhan (Project No. GPMR202201) to L. S. D. Rio Tinto Group are thanked for providing samples to D. F. The John de Laeter Centre acknowledges operational funding for the GeoHistory Facility from AuScope (auscope.org.au) and the Australian Government via the National Collaborative Research Infrastructure Strategy (NCRIS). The Australian Research Council is thanked for funding the Tescan Clara FESEM (ARC LE190100176), Tescan Integrated Mineral Analyser (ARC LE1400100150), the Selfrag HV pulse fragmentor (ARC LE130100219) and Laureate Fellowship grant to Z. X. L. (ARC FL150100133). M. T. D. W. publishes with permission of the Executive Director of GSWA. This is a contribution to IGCP 648: Supercontinent and Global Geodynamics.

## Author contributions

H.K.H.O. interpreted all analyses and led the drafting of the paper. D.F. and L.S.D. obtained funding and samples. Y.L. made the plate reconstructions. M.J.R. supplied the samples. M.D., D.J.C., N.J.E. and B.J.M. carried out the geochronological analyses. C.M. picked and imaged the diamond. B.I.A.M., A.L.J., Z.X.L., C.L.K. and M.T.D.W. supervised the work and provided geological and technical expertise. All authors contributed to editing of the manuscript.

## Competing interests

M.J.R. is an employee at Rio Tinto Group. The remaining authors declare no competing interests.
