## [Peer Review File · Nature Communications]

REVIEWER COMMENTS

Reviewer #1 (Remarks to the Author):

Olierook et al. Nature 2022

Lines 21 - 22. The Orapa kimberlite field and contained diamond mines (Orapa, Lethlakane, Damtshaa) lie within the Paleoproterozoic Makondi belt in Botswana. Similar to Argyle, in the Makondi belt, Paleoproterozoic crustal rocks are underlain by lithospheric mantle of Archean and/or Paleoproterozoic age. This might actually provide a more interesting take on the manuscript i.e., what other diamond mines are hosted by Paleoproterozoic crust +/- mantle (more than you think!).

Lines 22 - 23. The triggering mechanism is a very key concept that has been not satisfactorily discussed for lamproites, or carbonate rich olivine lamproites (aka CROL, formerly Gp II kimberlites, or orangeites). A number of mechanisms have been proposed to explain kimberlite melt triggers; they all appear valid i.e., there is not a single mechanism. See also Heaman et al (2003, 2004) for a discussion regarding opening of Iapetus and adjacent kimberlite, UML, and CROL at 650 – 550 Ma in Canada, Greenland, Baltica. Also opening of the South Atlantic wrt magmatic rocks of deep-seated origin is another possible example of this mechanism. Also note that kimberlites are not actually alkaline rocks (see your reference #2). There is another diamondiferous olivine lamproite field with past-producing mines (Ellendale) nearby; does your trigger mechanism work for Ellendale? Or is it different?

Broadening the discussion to other examples and 'similar' rock types to lamproites would provide a more global perspective, which needs to be done for a Nature paper.

Lines 37-50. See comments above re lines 21-23.

Line 48. high diamond grade drill core.

Line 52. Overview.

Line 67. maar-diatreme complex; or, a complex of four maar diatremes.

Lines 75-81. This section is a bit 'harsh' regarding previous work. In the supplemental data, note that 2 points is an errochron, not an isochron. Also, both Rb-Sr dates predate phlogopite leaching technique; combined these are perhaps a better reason for believing the date is potentially dubious.

The weighted mean (3 samples) K-Ar age of 1238 +/- 36 is similar to and has a significant overlap in uncertainty with the new titanite age of 1268 +/- 40. I. would have to suggest that the new titanite age does not represent a significant advance in the geochronology.

Line 81-83. Not obvious why triple quad Rb-Sr phlogopite dating was not undertaken on samples from the Argyle olivine lamproite dykes, to provide a potentially more robust age. Is the variation in all the available ages due to the fact that the diatremes and/or dykes are of different age?

Line 92 -97. A better explanation needed here. What is the fluid? What is the source of the fluid? Is the fluid related to the carbonate-rich fluid? There could be very little difference in age between lamproite emplacement, followed by carbonate alteration followed by 'hydrothermal' alteration.

Line 103. How was detrital apatite distinguished from magmatic apatite? Not clear.

Lines 145, 159. The minimum age could be as young as 1228 Ma....

Lines 182-184. Presumably there is some geochronology regarding Nuna breakup in this part of the Earth, which would be more useful than relying on paleomagnetic interpretations (that are based on geochronology).

Figure 4. Quite interesting; I like the application of a plate reconstruction. How valid is this reconstruction? Are there alternate re-constructions? Comment please.

From the perspective of a Nature paper – plot all global lamproite, kimberlite, and UML that are between the ages of 1350 – 1200 Ma (note there can be a lag between rifting and melt generation). Do you have a story? If yes, then I would think this is potentially a Nature paper.

Reviewer #3 (Remarks to the Author):

This is a welcome publication improving the age of the Argyle lamproite to be slightly older than previously thought. The quality of data, presentation of the content and significance of the finding warrants publication. I have two suggestions for consideration:

1. The role of supercontinent breakup in diamondiferous kimberlite genesis has been ventilated before (see Pandey and Chalapathi Rao ; Supercontinent transition as a trigger for ~ 1.1 Gyr diamondiferous kimberlites and related magmatism in India DOI: 10.1016/j.lithos.2020.105620) which need to be cited.
2. Kimberlites/lamproites coming under the age bracket of the newly obtained age of the Argyle lamproite should be compared on a global context since emplacement of such rocks is on a regional (continental) scale triggered by supercontinental breakups. For example, there are a number of lamproites in southern India (e.g., Chelima, Krishna) of this age bracket (1300-1400 Ma) which seems to have been missed. In many of the paleomagnetic fits NCC, India were together during that time scale.

Response to reviewers' comments

Legend:

Black text = reviewer's comment

Blue text = author's response

REVIEWER COMMENTS

Reviewer #1:

Olierook et al. Nature 2022

We thank the reviewer for their detailed comments to our manuscript. One thing to note is that this article was submitted to Nature Communications, rather than Nature.

Lines 21 - 22. The Orapa kimberlite field and contained diamond mines (Orapa, Lethlakane, Damtshaa) lie within the Paleoproterozoic Makondi belt in Botswana. Similar to Argyle, in the Makondi belt, Paleoproterozoic crustal rocks are underlain by lithospheric mantle of Archean and/or Paleoproterozoic age. This might actually provide a more interesting take on the manuscript i.e., what other diamond mines are hosted by Paleoproterozoic crust +/- mantle (more than you think!).

AGREE. We thank the reviewer for bringing the Orapa kimberlite field to our attention. We have nuanced the text to state that Argyle is "one of only a few" rather than the sole economic deposit in the world.

Lines 22 - 23. The triggering mechanism is a very key concept that has been not satisfactorily discussed for lamproites, or carbonate rich olivine lamproites (aka CROL, formerly Gp II kimberlites, or orangeites). A number of mechanisms have been proposed to explain kimberlite melt triggers; they all appear valid i.e., there is not a single mechanism. See also Heaman et al (2003, 2004) for a discussion regarding opening of Iapetus and adjacent kimberlite, UML, and CROL at 650 – 550 Ma in Canada, Greenland, Baltica. Also opening of the South Atlantic wrt magmatic rocks of deep-seated origin is another possible example of this mechanism.

AGREE. While we agree there are multiple mechanisms for kimberlites and lamproites, it has not been fully recognized that extension with (super)continental breakup is a significant cause. This is the main reason we chose to use the word 'under-recognised' in the last line of the abstract, as it's been documented before in case studies, but the holistic mechanism has not been appreciated. We thank the reviewer for the list of locations that probably share the same emplacement mechanism.

Also note that kimberlites are not actually alkaline rocks (see your reference #2).

AGREE. This particular sentence exclusively talks about the Argyle lamproite, which is alkaline. However, our final sentence in the abstract and conclusion does elude to kimberlites as well, and we removed 'alkaline' from these two sentences.

There is another diamondiferous olivine lamproite field with past-producing mines (Ellendale) nearby; does your trigger mechanism work for Ellendale? Or is it different?

PARTLY AGREE. While proximal to Argyle, Ellendale is vastly different, being a product of a Cenozoic plume tail migration across the Australian continent; it is dated at 20.6 +/- 2.8 Ma (Evans 2013, Mineralium Deposita). This is somewhat outside of the scope of this article.

Broadening the discussion to other examples and 'similar' rock types to lamproites would provide a more global perspective, which needs to be done for a Nature paper.

AGREE. While not strictly a Nature paper (rather Nature Communications), we agree that broadening the paper aids the international readership, and the references provided above and below are excellent. We have added three new paragraphs to the end of discussion to take broaden the scope of the paper, related to other examples of (super)continental breakup, proximity to Archean lithosphere and preservation bias.

Lines 37-50. See comments above re lines 21-23.

AGREE. Same response as the comment to L21-23.

Line 48. high diamond grade drill core.

AGREE. Fixed.

Line 52. Overview.

We are uncertain what the reviewer means here. It already states 'Overview' in the sub-section heading.

Line 67. maar-diatreme complex; or, a complex of four maar diatremes.

AGREE. Changed to maar-diatreme complex.

Lines 75-81. This section is a bit 'harsh' regarding previous work. In the supplemental data, note that 2 points is an errochron, not an isochron. Also, both Rb-Sr dates predate phlogopite leaching technique; combined these are perhaps a better reason for believing the date is potentially dubious. The weighted mean (3 samples) K-Ar age of 1238 +/- 36 is similar to and has a significant overlap in uncertainty with the new titanite age of 1268 +/- 40. I. would have to suggest that the new titanite age does not represent a significant advance in the geochronology.

PARTLY AGREE. We have been as objective as possible when critiquing the previous geochronology, but the accuracy of the previous analyses are simply not statistically rigorous. Either the analyses are overdispersed for a single population ($p < 0.05$) or, as the reviewer point out, a two-point line that cannot be statistically verified (any two points will always make a line!). Bob Pidgeon in his original article recognized that the phlogopite ages he was getting were minimum ages only due to the alteration of the grains; he even confessed last year in a discussion with us that he was "never quite happy with the age of Argyle".

Our new LA-ICP-MS-derived age of 1268 +/- 40 Ma is the first accurate age for Argyle, but we wholeheartedly agree that it is far from the most precise age. To this end, we have taken selected grains along the isochron that were previously analysed by LA-ICP-MS, plucked them out of resin, and dissolved them for high-precision ID-TIMS analyses at the British Geological Survey. Combined with the LA-ICP-MS data, this yields an age of 1257 ± 15 Ma, with a few data points slightly scattered, presumably due to alteration. As before, we suspect that this age of c. 1260 Ma is strictly a

minimum age as some post-emplacement alteration may still have affected the grains. This is observed in the ID-TIMS data and presumably present in the LA-ICP-MS data but masked by the higher uncertainties.

Line 81-83. Not obvious why triple quad Rb-Sr phlogopite dating was not undertaken on samples from the Argyle olivine lamproite dykes, to provide a potentially more robust age. Is the variation in all the available ages due to the fact that the diatremes and/or dykes are of different age?

AGREE. This was not explained in the text, but we have now added the following sentence:

“Moreover, whilst the excess radiogenic Ar issue may be obviated with in situ Rb-Sr dating^{17, 18}, minor alteration of the phlogopite and uncertainty whether the dykes are coeval with the diatreme still make dating these phlogopite grains problematic.”

Line 92 -97. A better explanation needed here. What is the fluid? What is the source of the fluid? Is the fluid related to the carbonate-rich fluid? There could be very little difference in age between lamproite emplacement, followed by carbonate alteration followed by ‘hydrothermal’ alteration.

AGREE. We have expanded this small section by including the following text that alludes to the fact that the titanite formation is probably within uncertainty of the lamproite emplacement:

“The titanite was likely formed by dissolving original perovskite from the lamproite with the addition of silica from the surrounding maar waters¹⁶, implying that the titanite was a product of deuteric alteration and only very shortly post-dates lamproite emplacement, well within uncertainty of any radiometric technique.”

Line 103. How was detrital apatite distinguished from magmatic apatite? Not clear.

This is currently stated as “Trace rounded zircon and subangular apatite are also present within the detrital matrix”, the implication being that there is no magmatic apatite.

Lines 145, 159. The minimum age could be as young as 1228 Ma....

PARTLY AGREE. The distribution of 1268 ± 40 Ma is Gaussian. In other words, there is no strict ‘minimum age’, but rather there is 95% confidence that the age is older than 1228 Ma. That being said, it is still more likely that the age is closer to 1268 Ma given the nature of a normal distribution. And with a new minimum age of 1257 ± 15 Ma, this is far more tightly bracketed.

Lines 182-184. Presumably there is some geochronology regarding Nuna breakup in this part of the Earth, which would be more useful than relying on paleomagnetic interpretations (that are based on geochronology).

AGREE. Yes and no. While there are plenty of rocks that can be dated between 1320 and 1220 Ma, almost all the well-dated samples with paleomagnetic data come from Laurentia, with a few at 1250 Ma from Antarctica and Brazil (see e.g. Ernst et al., 2008 Global record of 1600–700 Ma Large Igneous Provinces (LIPs): Implications for the reconstruction of the proposed Nuna (Columbia) and Rodinia supercontinents). Even though this article is now 15 years old, virtually no paleomagnetic data from other continents across this time frame exists. All that is clear is that at c. 1320 Ma Nuna was together, and at 1220 Ma it was dispersed. Australia has rocks that can be reliably dated and their paleomagnetic data collected at 1320 Ma (McArthur Basin, N. Australia) and 1220 Ma (Marnda Moorn LIP, SW Australia). It’s clear from Australia’s position that it’s broken away from Laurentia

sometime in the intervening 100 Myr. To illustrate this point, we have included the following text in the first paragraph of the discussion:

“Temporally-constrained paleomagnetic and geological evidence indicates that Nuna was together at c. 1320 but dispersed by c. 1220 Ma, with scarce high-quality paleomagnetic in the intervening 100 Myr meaning it is difficult to be more precise on the exact timing of breakup²⁸.”

Figure 4. Quite interesting; I like the application of a plate reconstruction. How valid is this reconstruction? Are there alternate re-constructions? Comment please.

AGREE. There are no other full-plate palinspastic reconstructions at this time (i.e., including oceanic crust). However, there are many others, with subtle differences between them based on different interpretations of paleomagnetic data and geological evidence (e.g., dyke swarms, orogenic belts). To incorporate these slight differences of interpretations, we’ve included the following at the start of the discussion:

“At c. 1300 Ma, the supercontinent Nuna was on the cusp of breakup²⁶. At the time, most plate tectonic reconstructions place Argyle at the periphery of the McArthur-Yanliao Gulf (Fig. 4), an intercontinental Mesoproterozoic basin between proto-Australia, North China, Siberia and Laurentia^{27, 28, 29, 30, 31, 32} (and in some reconstructions, India^{29, 31} and/or South China^{28, 29}).

From the perspective of a Nature paper – plot all global lamproite, kimberlite, and UML that are between the ages of 1350 – 1200 Ma (note there can be a lag between rifting and melt generation). Do you have a story? If yes, then I would think this is potentially a Nature paper.

AGREE. There are virtually no volatile- and incompatible element-rich ultramafic rocks that intruded between 1350 and 1200 Ma according to the compilations of Larry Heaman and Sebastian Tappe (and co-workers). There’s a few shoddy 1.3 Ga ages for Martin’s Drift and Helpmekaar (but the original publication place their original eruptions as likely older than 1.6 Ga). And there’s a whole host of magmatic products that kick off after 1220 Ma. Here’s our reasoning for this phenomena, now stated in the text:

“The implication is that continental extension is one of the dominant driving forces in ensuring that volatiles can be destabilized in the deep mantle and propagate towards the surface. That being said, there is still likely significant erosion of the breakup-related diatremes, and it is perhaps for this reason that diatremes concomitant with initial continental breakup is rare in the geological record. One only has to look at Argyle to appreciate the fortuity involved in preserving a diamond-bearing diatreme. It was buried to >5–6 km (> 180°C, zircon [U-Th]/He) and exhumed to <2 km by the Early Cretaceous, to only be partly eroded and at mineable depth at the present day (Fig. 1, Supplementary Information).”

Reviewer #3:

This is a welcome publication improvising the age of the Argyle lamproite to be slightly older than previously thought. The quality of data, presentation of the content and significance of the finding warrants publication. I have two suggestions for consideration.

We thank the reviewer for their comments on our paper.

1. The role of supercontinent breakup in diamondiferous kimberlite genesis has been ventilated before (see Pandey and Chalapathi Rao ; Supercontinent transition as a trigger for ~ 1.1 Gyr diamondiferous kimberlites and related magmatism in India DOI: 10.1016/j.lithos.2020.105620) which need to be cited.

AGREE. We have expanded this sub-section to the following:

"Lithospheric extension in or adjacent to Archean cratons during continental rifting has previously been recognized as a mechanism for emplacement of kimberlites, lamproites and other related volatile- and incompatible element-enriched ultramafic diatremes^{38, 39, 40, 41}. Examples include the East European Platform during Nuna breakup (c. 1250–1230 Ma⁴²), the Kimberley Craton during attempted Rodinia breakup (c. 840–800 Ma⁴³) and the Kaapvaal Craton during Gondwana/Pangea breakup (c. 200–120 Ma^{44, 45}), amongst many others. However, the role of (super)continental breakup has not been fully appreciated despite the prevalence of diatreme emplacements that occurred during periods of continental breakup^{2, 38}. In all breakup-related cases, there appears to some diatreme emplacement contemporaneous with breakup, but with a peak production of volatile-rich ultramafic rocks that lags the breakup period⁴¹. The same applies to Argyle and Nuna breakup. Whilst Argyle appears to be coincident with the onset of Nuna breakup, potentially together with late-stage products at Helpmekaar and Martin's Drift in South Africa⁴¹ and kimberlites in the East European platform⁴², it is not until 1200–1100 Ma that diatreme emplacement peaked worldwide, coincident with accelerated plate velocities³⁰. Thus, there appears to be a first order control of continental extension and plate velocities on (diamondiferous) kimberlite and related volatile-rich ultramafic diatreme production⁴¹."

2. Kimberlites/lamproites coming under the age bracket of the newly obtained age of the Argyle lamproite should be compared on a global context since emplacement of such rocks is on a regional (continental) scale triggered by supercontinental breakups. For example, there are a number of lamproites in southern India (e.g., Chelima, Krishna) of this age bracket (1300-1400 Ma) which seems to have been missed. In many of the paleomagnetic fits NCC, India were together during that time scale.

PARTLY AGREE. We have compiled known occurrences between c. 1310 and 1220 Ma (the range permissible for Argyle and Nuna breakup), including those that are diamond-bearing. However, the southern Indian intrusions are either older than c. 1350 Ma (both Chelima and Krishna) or younger than 1220 Ma, and are outside of the scope of this article.

For the paleomagnetic fits and alternative plate reconstructions, we have amended the text in the discussion to:

"At the time, most plate tectonic reconstructions place Argyle at the periphery of the McArthur-Yanliao Gulf (Fig. 4), an intercontinental Mesoproterozoic basin between proto-Australia, North China, Siberia and Laurentia^{27, 28, 29, 30, 31, 32} (and in some reconstructions, India^{29, 31} and/or South China^{28, 29})."

However, we do note that the work of Pisarevsky et al., 2013 (Palaeomagnetic, geochronological and geochemical study of Mesoproterozoic Lakhna Dykes in the Bastar Craton, India: Implications for the Mesoproterozoic supercontinent) shows that, at c. 1466 Ma, India and North China are likely not close to each other (and rather attached to Baltica).

REVIEWERS' COMMENTS

Reviewer #1 (Remarks to the Author):

Thanks for incorporating some of my suggestions

Reviewer #3 (Remarks to the Author):

I am happy to see the revised manuscript has addressed all the concerns raised by the reviewers. It looks certainly more focused and crisp.

Olierook, Nature Communications, 2nd review

1.) Revised keywords suggestions:

rifting (and/or LIP), Nuna supercontinent breakup, olivine lamproite, diamond mine

NB Derim Derim and Yanlia are too specific

2.) The 'lag' from the end of the LIP at 1300 - 1290 is 43 to 18 Myr. reasonable?

3.) Is the trigger (rifting) providing heat only, or some ubasic/melt + heat. What does the petrology say?

Line 45. olivine lamproite

Line 47. in non-Archean terranes

Line 103. maar-derive meteoric waters

Line 151. Sentence not completed

Line 182. and in some reconstructions also including, India

Line 208. emplacement age of the Argyle lamproite

Line 220. together with temporally near-coincident magmatism at Helpmekaar

NB Martin's Drift (Lerala) is 1364 Ma Chelima is 1354 Ma and both would be LIP/rifting precursor by 20-30Myr which is feasible, but from a geographic perspective these localities are on the other side of the globe so not likely.

Response to reviewers' comments for NCOMMS-22-47462A

Legend:

Black text = reviewer's comment

Blue text = author's response

REVIEWER COMMENTS

Reviewer #1:

Thanks for incorporating some of my suggestions

We thank the reviewer for their comments on our paper.

1.) Revised keywords suggestions:

rifting (and/or LIP), Nuna supercontinent breakup, olivine lamproite, diamond mine. NB Derim Derim and Yanlia are too specific

AGREE. We have updated these to:

- Rifting
- Nuna supercontinent breakup
- olivine lamproite
- diamond mine
- Halls Creek Orogen
- large igneous province

2.) The 'lag' from the end of the LIP at 1300 - 1290 is 43 to 18 Myr. reasonable?

AGREE. Certainly, and in line with recent research from a paper in last week's Nature issue (Gernon et al., 2023, *Rift-induced disruption of cratonic keels drives kimberlite volcanism*) that indicates a common 30 Myr lag followed breakup. We've now cited some of their new work.

3.) Is the trigger (rifting) providing heat only, or some ubasic/melt + heat. What does the petrology say?

Just heat, although a change in the oxidation state of the subcontinental lithospheric mantle may also play a role as Stephen Foley's work suggests. Argyle is situated in a failed Paleoproterozoic rift and did not fully reactivate in the Mesoproterozoic. Thus, it's unlikely that asthenospheric mantle experienced decompression melting during the Mesoproterozoic. The simpler explanation is that hydrated ultramafic rocks are prone to melting given any thermal driver, and that's why the olivine lamproites were emplaced shortly after continental breakup.

Line 45. olivine lamproite

AGREE. Changed.

Line 47. in non-Archean terranes

AGREE. Changed.

Line 103. maar-derive meteoric waters

AGREE. Changed.

Line 151. Sentence not completed

AGREE. Completed this sentence to:
“Potential causes for this scatter include recent Pb-loss through alteration *or* ancient Pb-loss during a subsequent tectonothermal event.”

Line 182. and in some reconstructions also including, India

AGREE. Changed.

Line 208. emplacement age of the Argyle lamproite

DISAGREE. We prefer to omit the term ‘age’ as it’s superfluous.

Line 220. together with temporally near-coincident magmatism at Helpmekaar NB Martin's Drift (Lerala) is 1364 Ma Chelima is 1354 Ma and both would be LIP/rifting precursor by 20-30Myr which is feasible, but from a geographic perspective these localities are on the other side of the globe so not likely.

AGREE. Changed.

Hi - sorry for the delay for the re-review - on holidays.

No problem – so was the lead author!

Reviewer #3:

I am happy to see the revised manuscript has addressed all the concerns raised by the reviewers. It looks certainly more focused and crisp.

We thank the reviewer for their comments on our paper.